# The ADAMTS/Fibrillin Connection: Insights into the Biological Functions of ADAMTS10 and ADAMTS17 and Their Respective Sister Proteases

**DOI:** 10.3390/biom10040596

**Published:** 2020-04-12

**Authors:** Stylianos Z. Karoulias, Nandaraj Taye, Sarah Stanley, Dirk Hubmacher

**Affiliations:** Orthopaedic Research Laboratories, Leni & Peter W. May Department of Orthopaedics, Icahn School of Medicine at Mount Sinai, New York, NY 10029, USA; stylianoszafeirios.karoulias@mssm.edu (S.Z.K.); nandaraj.taye@mssm.edu (N.T.); sarah.stanley@icahn.mssm.edu (S.S.)

**Keywords:** extracellular matrix, ADAMTS proteases, fibrillin, microfibrils, Weill-Marchesani syndrome, short stature, lens dislocation

## Abstract

Secreted a
disintegrin-like and metalloprotease with thrombospondin type 1 motif (ADAMTS) proteases play crucial roles in tissue development and homeostasis. The biological and pathological functions of ADAMTS proteases are determined broadly by their respective substrates and their interactions with proteins in the pericellular and extracellular matrix. For some ADAMTS proteases, substrates have been identified and substrate cleavage has been implicated in tissue development and in disease. For other ADAMTS proteases, substrates were discovered in vitro, but the role of these proteases and the consequences of substrate cleavage in vivo remains to be established. Mutations in *ADAMTS10* and *ADAMTS17* cause Weill–Marchesani syndrome (WMS), a congenital syndromic disorder that affects the musculoskeletal system (short stature, pseudomuscular build, tight skin), the eyes (lens dislocation), and the heart (heart valve abnormalities). WMS can also be caused by mutations in fibrillin-1 (*FBN1*), which suggests that ADAMTS10 and ADAMTS17 cooperate with fibrillin-1 in a common biological pathway during tissue development and homeostasis. Here, we compare and contrast the biochemical properties of ADAMTS10 and ADAMTS17 and we summarize recent findings indicating potential biological functions in connection with fibrillin microfibrils. We also compare ADAMTS10 and ADAMTS17 with their respective sister proteases, ADAMTS6 and ADAMTS19; both were recently linked to human disorders distinct from WMS. Finally, we propose a model for the interactions and roles of these four ADAMTS proteases in the extracellular matrix.

## 1. The ADAMTS Protease Family

The family of a
disintegrin-like and metalloprotease with thrombospondin type 1 motif (ADAMTS) comprises 19 secreted metalloproteases, which are primarily involved in the formation, remodeling, and/or degradation of components of the extracellular matrix (ECM) [1]. All of the ADAMTS proteases share a similar domain organization with a conserved N-terminal protease domain and a variable C-terminal ancillary domain, which is implicated in substrate recognition and ECM binding (Figure 1a) [2,3,4]. The conserved ADAMTS protease domain consists of a signal peptide to target ADAMTS proteases for secretion, a propeptide that is typically removed by furin/PACE proprotein convertases to activate ADAMTS proteases, the catalytic metalloproteinase domain itself, and a disintegrin-like domain. The ancillary domain, which mediates binding to ECM components and specific protease substrates, consists of a conserved central thrombospondin type 1 sequence repeat motif (TSR), a cysteine-rich domain, and a spacer domain. The subsequent C-terminal domains and their arrangement distinguish individual ADAMTS proteases and can comprise combinations of up to 14 TSRs, CUB domains, a mucin/proteoglycan domain, or the GON-1 domain [5].

The individual proteases of the ADAMTS family can be grouped according to their evolutionary homology and their biological substrates, e.g., proteoglycanases (ADAMTS1, 4, 5, 8, 9, 15, and 20) or procollagen N-propeptidases (ADAMTS2, 3, and 14). Until about a decade ago, no substrates were reported for the ADAMTS proteases that form the central clade of the ADAMTS homology tree, i.e., ADAMTS6, 7, 10, 12, 16, 17, 18, and 19, and these ADAMTS proteases were considered “orphan” [6,7,8]. However, more recent work identified several substrates for these proteases, many of them are linked to fibrillin biology, such as fibrillin-1 and fibrillin-2 themselves, several latent transforming growth factor (TGF) β binding proteins (LTBPs), fibronectin, or ADAMTS-like 6 [9,10,11,12,13]. The consilience between evolutionary conservation and functionally related ECM substrates suggests that at least some of these ADAMTS proteases perform biological roles related to the formation or function of fibrillin microfibrils [14]. Fibrillin microfibrils represent pivotal ECM signaling platforms integrating TGFβ, bone morphogenetic protein (BMP), and mechanosignaling [15]. The fact that mutations in *ADAMTS10*, *ADAMTS17*, fibrillin-1 (*FBN1*), *LTBP2*, or *LTBP3* can cause almost identical short stature syndromes, called acromelic dysplasias, further supports the concept that some ADAMTS proteases in this central clade, specifically ADAMTS10 and ADAMTS17, are functionally connected through their ECM substrates [14,16,17,18]. Here, we review the recent literature on the homologous protease pairs ADAMTS6/ADAMTS10 and ADAMTS17/ADAMTS19. We explore the connection of ADAMTS10 and ADAMTS17 to fibrillin microfibril biology based on the consilience in human genetic disorders, sequence homology, and experimental evidence, and we develop a conceptual model of how these proteases may interact and cooperate in the pericellular matrix (PCM) and the ECM. We also include a discussion of the respective sister proteases, ADAMTS6 and ADAMTS19, since it is known for other ADAMTS protease pairs that they can cooperate or functionally compensate for each other during tissue development or in tissue homeostasis [19,20,21].

## 2. Domain Organization and Posttranslational Modifications of ADAMTS6, 10, 17, and 19

On the protein level, ADAMTS6, 10, 17, and 19 share the same domain organization (Figure 1a). However, each protease pair arose from distinct gene duplication events [30]. When comparing the nucleotide and amino acid sequences between the four proteases, it is evident that ADAMTS10 sequences are more similar to ADAMTS6, where 60% of the amino acid residues are identical, and ADAMTS17 sequences are more similar to ADAMTS19, with 56% of the amino acid residues being identical (Figure 1b) [1,31]. Despite the evolutionary homology, the identical domain organization, and the relatively high amino acid sequence identity, the ADAMTS proteases that form the individual protease pairs, ADAMTS6/ADAMTS10 and ADAMTS17/ADAMTS19, appear to have distinct biological functions, based on their involvement in different human disorders (see below). One possible explanation for the diversification in the function of these proteases could be differences in posttranscriptional and posttranslational modifications. 

Alternative splicing is a posttranscriptional mechanism that can expand the diversity and thus function of ADAMTS proteases by generating different isoforms. ADAMTS proteases have several splice variants based on the NCBI protein database. There are 13 isoforms listed for ADAMTS6, 4 for ADAMTS10, 12 for ADAMTS17, and 5 for ADAMTS19. For most of these isoforms, tissue-specific expression or functional data are not available. However, by homology mapping with ADAMTS10 as a template and analysis of expressed sequence tags in the GenBank™ database the existence of at least two splice variants for ADAMTS6 were predicted and subsequently shown experimentally in epithelial cells [32,33]. In addition, northern blot analysis of total mRNA isolated from adult human tissue demonstrated two ADAMTS10 mRNA species that differed in size, suggesting alternative splicing of ADAMTS10 mRNA as well [33]. Two isoforms of ADAMTS17 with distinct expression patterns have been described previously [25]. Our own unpublished data show expression of at least three additional isoforms of ADAMTS17 that differ in the sequence of the spacer domain (Balic, et al., manuscript in preparation).

In addition to alternative splicing, ADAMTS6, 10, 17, and 19 show differences in the number and location of predicted and experimentally verified posttranslational modifications, such as furin/PACE-processing, autocatalysis, N-glycosylation, or O-fucosylation. Based on western blot analysis, ADAMTS6 and ADAMTS19 are furin-processed but do not undergo apparent autocatalysis (Karoulias et al., unpublished data for ADAMTS19) [34]. However, a direct comparison between active ADAMTS6 and an inactive mutant form was not shown to completely rule out the possibility of ADAMTS6 autocatalysis. The propeptide of ADAMTS17 is also processed by furin, but in contrast to ADAMTS6 and ADAMTS19, ADAMTS17 undergoes extensive autoproteolysis at the cell surface or in the ECM [13]. Furin-processing of the ADAMTS17 propeptide is not required to activate ADAMTS17. Instead, the ADAMTS17 propeptide may act as a chaperone to facilitate ADAMTS17 secretion, since removal of the propeptide did abolish ADAMTS17 secretion or its release from the cell surface [13]. A similar role was previously described for the propeptide of ADAMTS9 [35]. ADAMTS10 on the other hand has a degenerated consensus sequence for furin-processing (GLKR instead of RLKR) and thus the propeptide remains covalently associated with the ADAMTS10 protease after secretion. Wild type ADAMTS10 has weak protease activity against fibrillin-1 [36]. However, upon restoration of the consensus furin-processing site in recombinant ADAMTS10, the propeptide was efficiently excised and proteolytic activity of ADAMTS10 against fibrillin-1 was enhanced [36].

ADAMTS proteases can be N-glycosylated and O-fucosylated. ADAMTS6 and ADAMTS10 contain six predicted N-glycosylation sites but their location is different (Figure 1a and Table 1). For example, four of the six predicted N-glycosylation sites in ADAMTS6 are located in the propeptide domain while ADAMTS10 harbors only two of the six predicted N-glycosylation sites in the propeptide. Interestingly, one of the predicted N-glycosylation sites in ADAMTS10 is located in the catalytic domain and it is tempting to speculate that this site may modulate proteolytic activity if differentially glycosylated. ADAMTS17 has seven predicted N-glycosylation sites with five of them being located in the ancillary domain. In contrast, ADAMTS19 contains only five predicted N-glycosylation sites. In addition to N-glycosylation, TSR domains of ADAMTS proteases can undergo O-fucosylation at the Cxx(S/T)C consensus motif as part of a non-canonical quality control pathway in the endoplasmic reticulum [37,38]. Three of the five TSR domains of ADAMTS6 and ADAMTS10 and four of the five TSR domains of ADAMTS17 and ADAMTS19 contain the consensus sequence for O-fucosylation [39]. For ADAMTS17, it was shown that TSR1, 3, and 5 were fully modified [13]. TSR4 was not O-fucosylated, despite the presence of the consensus motif. Functionally, the absence of O-fucosylation on TSR3 and TSR5, and to a lesser extent on TSR1, resulted in the lack of secretion of recombinant ADAMTS17 [13]. It is anticipated that O-fucosylation of TSRs in ADAMTS6, 10, and 19 would play a similar role in protein quality control and protein secretion. However, experimental data is lacking and the degree of O-fucosylation needs to be determined for each of these proteases.

Taken together, different isoforms of ADAMTS6/ADAMTS10 and ADAMTS17/ADAMTS19 proteases, generated by alternative splicing or differential posttranslational modifications, could be important determinants for the regulation of ADAMTS secretion or cell surface localization, for ADAMTS activation or modulation of protease activity, or for substrate recognition and ECM binding, and thus may provide the molecular underpinnings for differences in their biological functions.

## 3. Gene Expression Patterns of ADAMTS6, 10, 17, and 19

The biological functions of ADAMTS6, 10, 17, and 19 are not only determined by differential posttranscriptional and posttranslational modifications, which may guide ECM localization and substrate binding and cleavage, but also by the cell types and tissues that express these proteases. *Adamts10* for example is widely expressed in most adult tissues in mice and almost universally expressed during mouse embryonic development with the exception of the ectoderm [11,33]. RNA in situ hybridization demonstrated dynamic *Adamts10* expression, which was generally low during early embryonic development [embryonic day (E) 9.5 to E12.5] and induced in later stages of development (E14.5 and E17.5) [33]. At E14.5, *Adamts10* was expressed in several craniofacial tissues, notably in the perichondrium and periosteum, but not in cartilage, of newly forming bones of the mandible, and in the tongue muscle. In developing lungs, *Adamts10* was expressed in the connective tissue between the bronchi and the accompanying blood vessels. *Adamts10* was also expressed in the stomach, duodenum, pancreas, dorsal root ganglia, and primary ossification centers of vertebrae, but was absent in the liver. In the developing musculoskeletal system, *Adamts10* mRNA was detected between the cartilaginous metacarpals and metatarsals of the hand and feet and in dense connective tissues such as the joint capsule and developing tendons and ligaments. At E17.5 *Adamts10* expression was strong in the cartilage of developing bones and in the wall of large arteries. *Adamts10* was expressed in several ocular tissues during development, including non-pigmented ciliary epithelium, lens fiber cells, and parts of the retina [11]. Expression of ADAMTS10 in the eye is relevant for the characteristic ectopia lentis phenotype observed in individuals with Weill–Marchesani syndrome (WMS). All these data support a role for ADAMTS10 in the early steps of tissue and organ formation. *Adamts10* continued to be expressed in adult chondrocytes, tendon, and skeletal muscle in mice [11]. In human adult tissue, *ADAMTS10* mRNA was expressed in the heart, brain, lung, pancreas, but *ADAMTS10* expression was notably low or absent in skeletal muscle [33]. Given the broad expression of ADAMTS10 during embryonic development, it was somewhat surprising, that most *Adamts10* knockout mice apparently undergo normal development, with limited embryonic lethality (see below) [10,11].

Similar to *ADAMTS10*, *ADAMTS17* is widely expressed in many fetal and adult tissues in humans [25]. Two isoforms of *ADAMTS17*, isoform a (22 exons) and isoform b (16 exons), with distinct expression patterns, were identified. Both *ADAMTS17* isoforms were expressed in lung, brain, and the eye, including the retina. Immunostaining in human normal and cancer tissues showed widespread expression in multiple tissues to varying degrees [25]. However, no images were shown and the reported localization of ADAMTS17 in the cytoplasm raises the possibility that the antibody used in this study was non-specific in tissue staining, even though it recognizes recombinant ADAMTS17 protein [13]. In-situ hybridization studies in mouse embryonic tissue at E16.5 and in neonates demonstrated *Adamts17* expression in the eye, especially at the lens equator and in the trabecular region [13]. Apart from the eye, *Adamts17* mRNA was detected in the perichondrium, the intervertebral disc, and in the smooth muscle cell layer of blood vessel walls. The most pronounced expression of *Adamts17* was observed in lung parenchyma, skin epidermal basal cell layer, and in developing hair follicles. *Adamts17* mRNA was notably absent from the growth plate of long bones, the endothelium of blood vessels, and the bronchial epithelium in lung tissue.

Less detailed information about the expression patterns of *ADAMTS6* and *ADAMTS19* is currently available. *ADAMTS6* mRNA was detected in the mouse heart, specifically in the outflow tract, the heart valves, the atria, and the ventricular myocardium [34]. In northern blot analysis, *ADAMTS6* mRNA was weakly expressed in placental tissue, but was not detected in mouse embryonic development or in adult human tissues [40]. *ADAMTS19* mRNA was detected in fetal lung and osteosarcoma tissue [41]. More recently, *Adamts19* was shown to be expressed in the heart valve at all stages of heart valve formation and elongation and in bone and cartilage anlagen at E10.5 [42].

Taken together, *ADAMTS10* and *ADAMTS17* show broad, both overlapping and distinct gene expression patterns in tissues. Therefore, it is possible that ADAMTS10 and ADAMTS17 may compensate for each other in mouse knockout models of the individual proteases at least in some tissues (see below). If these proteases can indeed compensate for each other in murine loss-of-function models, similar to what has been observed for the ADAMTS7/ADAMTS12 and ADAMTS9/ADAMTS20 pairs, needs to be explored [19,20]. No compensation of *ADAMTS10* with *ADAMTS17* or vice versa appears to occur in tissues affected in WMS, specifically the eye and the musculoskeletal system. However, a lack of cardiac involvement and joint contractures in WMS4 due to *ADAMTS17* mutations leaves the possibility that compensation with *ADAMTS10*, but not vice versa, causes the lack of cardiac and joint presentations in WMS4 (see below). *ADAMTS6* and *ADAMTS19* appear to have more tissue-specific expression patterns, which are reflected in the specific diseases associated with mutations in *ADAMTS6* and *ADAMTS19,* as described in the next section.

## 4. Disorders Associated with Mutations in *ADAMTS6, 10, 17,* and *19*

Mutations in *ADAMTS10* and *ADAMTS17* cause WMS [16,25,26,27]. WMS is characterized by severe short stature and brachydactyly in combination with lens dislocation, thickened skin, and joint contractures. However, frequent cardiac involvement is reported for WMS1 caused by mutations in *ADAMTS10*, but is typically not observed in individuals with WMS4 due to *ADAMTS17* mutations. Similarly, joint stiffness is underrepresented in WMS4. Given that there are only a few individuals with WMS reported worldwide, definitive genotype–phenotype correlations are currently not feasible. Therefore, it is unclear if WMS caused by mutations in *ADAMTS10* or *ADAMTS17* represent different ends of a spectrum of WMS severity or if WMS due to mutations in *ADAMTS10* or *ADAMTS17* represent distinct disease entities, as suggested by the prior use of “WMS-like syndrome” for WMS4 [25]. *ADAMTS10* and *ADAMTS17* mutations causing WMS are spread out over the entire protein with no specific mutation hot spots (Figure 1a). However, most mutations are located in the domains comprising the catalytic domain and there is a notable paucity of mutations located in the ancillary domains. Mutations in *ADAMTS17* are also associated with short stature, lens dislocation, and glaucoma in several dog breeds, suggesting a fundamental role for ADAMTS17 in bone growth and in the formation and/or homeostasis of the ciliary zonule which is comprised of fibrillin microfibrils and holds the lens in place [43,44,45,46,47]. *ADAMTS10* and *ADAMTS17* were identified in a genome-wide association study (GWAS) of more than 12,000 cases versus 390,000 controls as susceptibility loci likely to cause carpal tunnel syndrome [48]. In addition, *ADAMTS10* and *ADAMTS17* were identified in a GWAS for genes influencing differential body fat distribution in women compared to men [49]. The individual contributions of *ADAMTS10* or *ADAMTS17* to traits governed by many genes are expected to be small. However, the possibility of studying double knockout mice for ADAMTS10 and ADAMTS17 may provide an opportunity to gain some insights into the contribution of these protease to multigenic traits, such as carpal tunnel syndrome or body fat distribution. 

Mutations in *ADAMTS6* are associated with altered ventricular conduction in the heart and cause a prolonged QRS interval observed by electrocardiography [34]. A prolonged QRS interval is an independent predictor for mortality in patients with cardiac disease as well as for the general population. Mutations in *ADAMTS19* cause non-syndromic heart valve disease [42]. 

In functional studies, disease-associated point mutations in recombinant ADAMTS6, 10, or 17 collectively abolished or reduced secretion of these proteases [16,26,34]. Therefore, WMS, prolonged QRS syndrome, and heart valve disease could arise from the absence of the proteases from the cell surface or the ECM, or from the acquisition of a dominant negative function due to the aberrant retention of the mutated ADAMTS proteases in the secretory pathway. ADAMTS19 mutations have not been analyzed in vitro.

## 5. Mouse Models of ADAMTS6, 10, 17, and 19 Deficiency

Phenotypes of knockout mice have been reported for *Adamts6*, *Adamts10*, *Adamts17*, and *Adamts19* [10,11,34,42,50]. The *Adamts6* knockout was identified in an ethylnitrosourea (ENU) mutagenesis screen for congenital heart defects [34]. *Adamts6*-deficient mice show developmental heart defects, including double outlet right ventricles, atrioventricular septal defects, and ventricular hypertrophy. *Adamts6*-deficient mice die at or around birth.

Two different mouse strains targeting *Adamts10* were reported that show overlapping and, somewhat surprising, also distinct phenotypes. In the first strain, 41 nucleotides in exon 5 of *Adamts10* were deleted via homologous recombination resulting in a reading frame shift and in presumed nonsense-mediated ADAMTS10 mRNA decay [11]. These *Adamts10* knockout mice showed some embryonic lethality in a pure C57BL/6 background which was not observed after backcrossing in a mixed C57BL/6 × 129/Sv background. *Adamts10* knockout mice showed reduced weight gain, but no abnormalities in the musculoskeletal system that would be characteristic for WMS. ADAMTS10-deficient eyes developed a mild shallowing of the anterior chamber, indicative of a potential abnormal positioning of the lens. However, the ciliary zonule was intact. Despite the presence of an intact ciliary zonule, immunostaining showed that fibrillin-2, which is normally not detected in the adult ciliary zonule, persisted beyond prenatal and early postnatal development, suggesting that ADAMTS10 may play a role in the degradation or masking of fibrillin-2 in the postnatal ciliary zonule [11,51]. However, the consequences of persistent fibrillin-2 microfibrils for the function of the ciliary zonule remain to be established. In the second mouse model, a human WMS1-causing *ADAMTS10* mutation (R237X) was introduced at the homologous location in the mouse *Adamts10* gene (S236X) using CRISPR-Cas9 genome editing [10]. The mutation did not result in nonsense-mediated mRNA decay, but resulted in the expression of a truncated form of ADAMTS10, essentially comprising the propetide domain. In addition to a reduction in weight gain found in both *Adamts10* models, homozygous *Adamts10*-S236X mice developed hallmark features of WMS, i.e., shorter long bones, thicker skin, and increased muscle mass. The long bones were 6%–10% shorter in ADAMTS10-deficient mice compared to wild type controls and alterations in the growth plate, i.e., a shorter resting zone with increased proliferation and an extended hypertrophic zone, were observed. The increased skeletal muscle mass in *Adamts10*-S236X knockout mice was attributed to the presence of more, but smaller myofibers compared to controls. The eyes of *Adamts10*-S236X mice had a smaller ciliary body with altered ciliary processes and showed a similar persistence of fibrillin-2 in the ciliary zonule of adult ADAMTS10-deficient eyes compared to the *Adamts10*-strain described above. Since both mouse strains show overlapping and distinct features, notably in the WMS-related musculoskeletal features which were present in *Adamts10*-S236X mice, it is tempting to speculate if the propeptide domain of ADAMTS10, which could be detected in the *Adamts10*-S236X mice, exerts a dominant negative effect contributing to the development of the musculoskeletal manifestations in *Adamts10*-S236X mice. This is especially relevant given the fact that the ADAMTS10 propeptide cannot be cleaved by furin and as such may not be released as a distinct peptide entity in wild type mice or humans.

An *Adamts17* knockout mouse was generated by Cre/Lox-mediated deletion of a portion of exon 4, which generates a frameshift mutation and results in ADAMTS17 mRNA depletion [50]. Similar to *Adamts10* knockout mice, *Adamts17* knockout mice showed some neonatal or perinatal lethality. *Adamts17* knockout mice showed features characteristic of WMS, including shorter long bones, brachydactyly, and thick skin. Bone shortening in *Adamts17* mice was also attributed to alterations in the growth plate. In contrast to *Adamts10*-S236X mice, a shortened hypertrophic zone in the growth plate of the tibia, with no alterations in chondrocyte proliferation, was observed in *Adamts17*-deficient mice. Mechanistically, canonical bone morphogenetic protein (BMP) signaling was attenuated in *Adamts17*-deficient primary chondrocytes. An eye phenotype was not reported for the *Adamts17* knockout mice.

Very recently, a cardiac valve phenotype was described in *Adamts19* knockout mice, recapitulating the heart valve phenotype in human non-syndromic heart valve disease caused by *ADAMTS19* mutations [42]. *Adamts19*-deficient mice develop thickened aortic valves that can result in aortic regurgitation and severe aortic valve stenosis. However, the cardiac valve phenotype was present in only ~40% of *Adamts19* knockout mice, suggesting partial compensation by other proteases, potentially ADAMTS10 or ADAMTS17. Expression of ADAMTS10 was indeed observed in the leaflets of the aortic valve and WMS1 due to *ADAMTS10* mutations can result in mitral valve prolapse [11,52].

## 6. Substrates and Proposed Biological Functions for ADAMTS6, 10, 17, and 19

Given that recessive mutations in *ADAMTS10* or *ADAMTS17* or dominant mutations in *FBN1* can all cause WMS, a role for both of these ADAMTS proteases in fibrillin microfibril formation and homeostasis was predicted and subsequently confirmed. Wild-type ADAMTS10 cleaves recombinant fibrillin-1 inefficiently, but fibrillin-1 proteolysis was significantly enhanced when the consensus furin site in recombinant ADAMTS10 was restored [36]. More recently, ADAMTS10 was shown to cleave fibrillin-2 in vitro, but again only after furin processing was enabled [11]. This indicates that ADAMTS10 has intrinsic protease activity and may act as a fibrillinase, but the question remains, if and how ADAMTS10 is activated in vivo. Alternatively, optimal conditions for maximum ADAMTS10 protease activity, even in the absence of furin processing, may be present in tissues and not be recapitulated in vitro. In addition, the intrinsic low fibrillin-1 cleavage activity in wild-type ADAMTS10 may be sufficient to mediate fibrillin microfibril turnover in vivo, given that remodeling and turnover of ECM proteins in tissues can be a slow process [53,54,55]. In two mouse models of ADAMTS10 deficiency, fibrillin-2 microfibrils were shown to persist in the ciliary zonule in the eye, supporting a proteolytic role for ADAMTS10 in the postnatal remodeling of the ciliary zonule, specifically in the turnover of fibrillin-2 [10,11]. ADAMTS10 has two fibrillin-1 binding sites and binds to the fibrillin-1 N- and C-terminus. The binding site for ADAMTS10 in the fibrillin-1 N-terminus was mapped to exons 1-11 and overlaps with the binding site for some ADAMTS-like proteins and many other fibrillin-1 binding proteins [36,56,57,58]. When recombinant, wild-type ADAMTS10, i.e., non-furin processed ADAMTS10, was added to human dermal fibroblasts, it promoted the deposition of fibrillin microfibrils and as such, ADAMTS10 may play a more prominent role in modulating fibrillin microfibril assembly rather than in fibrillin proteolysis in vivo [36]. The persistence of fibrillin-2 immunoreactivity in the postnatal ciliary zonule of ADAMTS10-deficient eyes may then be explained by a lack of fibrillin-1 microfibril formation, which was shown to co-assemble with fibrillin-1 and to mask fibrillin-2 microfibrils in postnatal tissues [59,60,61].

Much less is known about the molecular function and substrate spectrum of ADAMTS6, the sister protease of ADAMTS10. It was shown that ADAMTS6 binds to a similar N-terminal region of fibrillin-1 (exons 8–11) and that ADAMTS6 binding was not affected by a WMS mutation introduced in recombinant fibrillin-1 [56]. Overexpression of ADAMTS6 in epithelial cells did inhibit fibrillin-1 microfibril formation and also resulted in a reduction of fibronectin in the ECM. This was in contrast to ADAMTS10, which promoted fibrillin assembly in mesenchymal and epithelial cells. In addition, it was shown that ADAMTS6 can cleave LTBP1 and syndecan 4 (SDC4) [56]. However, the active site mutant form of ADAMTS6 was not used as an important negative control in these experiments. In contrast to ADAMTS10, ADAMTS6 bound strongly to a C-terminal region of LTBP1 and thus it may play a role in the regulation of TGFβ signaling. In summary, ADAMTS10 may promote microfibril formation independent of its proteolytic activity. Given the degenerated furin consensus sequence, ADAMTS10 may act similar to an ADAMTS-like protein while ADAMTS6 may work as a true protease. ADAMTS-like proteins are part of the ADAMTS family since they share homology with the ancillary domain of ADAMTS proteases, but lack the protease domain [1,14]. This is of relevance since several ADAMTS-like proteins are implicated in fibrillin microfibril formation and function, including ADAMTS-like 2, 4, 5, and 6 [62,63,64,65,66,67]. In addition, mutations in *ADAMTSL2* cause geleophysic dysplasia, which, like WMS, belongs to the acromelic dysplasia group of disorders which all share the musculoskeletal and skin phenotypes [68]. Mutations in ADAMTSL4 cause ectopia lentis (lens dislocation), which is a key feature of WMS, and the protein composition of the ciliary zonule was altered in ADAMTS10-deficienct eyes [63,69].

ADAMTS17 binds to fibrillin-1 microfibrils and recombinant fibrillin-2 but it did not cleave either of the fibrillin isotypes in vitro [13]. In contrast to ADAMTS10, ADAMTS17 is furin processed and undergoes extensive autoproteolysis resulting in the release of multiple ADAMTS17 peptides from the cell surface or from the ECM [13]. However, if autoproteolysis of ADAMTS17 occurs in vivo remains to be determined. When analyzing primary skin fibroblasts from patients with WMS due to *ADAMTS17* mutations, we observed a reduction in fibronectin and fibrillin-1 secretion and their deposition in the ECM [16]. These findings suggested that fibronectin or fibrillin-1 could be potential substrates for ADAMTS17 or that ADAMTS17 or its autoproteolytic peptides could play a role in the maturation or biogenesis of these ECM networks, independent of direct ADAMTS17 catalytic activity [16]. When WMS1 patient-derived fibroblasts harboring an *ADAMTS10* mutation were analyzed, fibrillin microfibril deposition was similarly reduced [36]. The observation that fibrillin-1 was co-retained in fibroblasts derived from individuals with WMS4 due to *ADAMTS17* mutations, would suggest a potential role for ADAMTS17 as a chaperone to facilitate fibrillin-1 secretion and possibly secretion of other ECM proteins [16]. When the WMS4 mutation was introduced in recombinant ADAMTS17, we observed the retention of mutant ADAMTS17 in the secretory pathway. As such, WMS mutations in ADAMTS17 could also exert a dominant negative effect, including non-canonical proteolysis of proteins in the secretory pathway.

In contrast to ADAMTS17, its sister protease ADAMTS19, undergoes furin processing, but does not undergo autoproteolysis (Karoulias et al., unpublished data). So far, no substrates for ADAMTS19 have been reported. Similar to the ADAMTS6 and ADAMTS10 pair, the possibility exists that ADAMTS17 may act more like an ADAMTS-like protein through its bioactive peptides and ADAMTS19 may be a true protease involved in proteolytic ECM remodeling in vivo.

## 7. Outlook: How ADAMTS6, 10, 17, and 19 May Cooperate in ECM Formation and Remodeling

So far, the spectrum of substrates for the ADAMTS6/ADAMTS10 and ADAMTS17/ADAMTS19 protease pairs and their involvement in biological and pathogenic mechanisms are only emerging. Some of the proposed interactions of these ADAMTS proteases and the open questions associated with processes in which these ADAMTS proteases might play a role, are summarized in Figure 2. ADAMTS10 is an inefficient protease (fibrillinase) in vitro due to the absence of furin-mediated activation, while ADAMTS17 undergoes furin processing and inactivates itself, due to autocatalysis. It is possible that both of these ADAMTS proteases work more similar to ADAMTS-like proteins and their biological function could be independent of their respective proteolytic activities. They could modulate ECM formation and function or modulate the function of other ADAMTS proteases by facilitating or preventing substrate binding through their ancillary domains. If ADAMTS10 works as a protease in vivo, the mechanism of ADAMTS10 activation or the conditions required for optimal protease activity need to be determined. Alternatively, it is possible that substrates that are cleaved efficiently by ADAMTS10 are not yet identified or that the low protease activity of ADAMTS10 in the absence of furin cleavage may be sufficient to mediate a slow but steady turnover of fibrillin microfibrils in tissues.

Given the fact that mutations in *ADAMTS10*, *ADAMTS17*, or *FBN1* result in WMS, one possibility is that ADAMTS17 activates ADAMTS10 when both proteases are sequestered to fibrillin microfibrils in the ECM. Fibrillin-1 binding could induce conformational changes in ADAMTS10 that would expose a putative cryptic cleavage site for ADAMTS17. The resulting proteolysis by ADAMTS17 would activate ADAMTS10 and convert the inactive enzyme into a more efficient protease. Alternatively, ADAMTS17 or its autocatalytic peptides could activate ADAMTS10 independent of propeptide processing through allosteric mechanisms modulating ADAMTS10 protein conformation. We are currently exploring the ADAMTS10-ADAMTS17-fibrillin-1 microfibril network experimentally with mouse models and with in-vitro systems. The connection of ADAMTS6 and ADAMTS10 is so far only studied in epithelial cells where ADAMTS10 suppresses the expression of ADAMTS6 and both ADAMTS proteases show opposing effects on fibrillin microfibril formation [56]. The functional relevance of that observation and an extension of that function to other cell types will reveal if there is indeed a functional interaction between ADAMTS6 and ADAMTS10. A double knockout of both proteases in mice could be very informative. Mutations in *ADAMTS19* were linked to a human disorder only very recently [42]. It will be exciting to see which ADAMTS19 substrates will be identified in the future and if there is functional overlap with ADAMTS17 or possibly ADAMTS6 or ADAMTS10, given the incomplete penetrance of the heart valve phenotype in *Adamts19* knockout mice. It will be equally exciting to see how the ADAMTS-fibrillin connections will be further disentangled and what the identification of novel substrates will suggest about the biological functions of this evolutionary related subgroup of ADAMTS proteases.

## Figures and Tables

**Figure 1 biomolecules-10-00596-f001:**
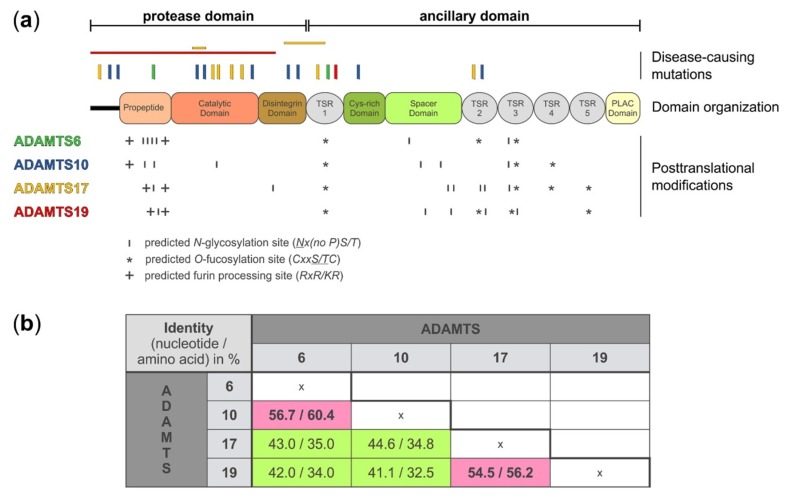
A disintegrin-like and metalloprotease with thrombospondin type 1 motif (ADAMTS)6, 10, 17, and 19 domain organization, location of disease-causing mutations, and sites for putative post-translational modifications. (**a**) Domain organization, depicting the protease and ancillary domain of ADAMTS6, 10, 17, and 19, which have an identical domain organization. However, the four ADAMTS proteases show differences in the number and localization of predicted sites for posttranslational modifications, such as N-glycosylation (http://www.cbs.dtu.dk/services/NetNGlyc/), O-fucosylation, and furin processing which could specify individual substrates, define protein-protein interactions, or govern autocatalytic properties that collectively distinguish these four ADAMTS proteases from each other. The location of disease-causing mutations is indicated on top of the diagram. Point mutations are indicated with vertical bars and deletion of larger gene fragments are indicated with horizontal bars [16,22,23,24,25,26,27,28,29]. The disease-causing mutations depicted for each ADAMTS protease are color-coded to match the color of the individual ADAMTS protease in panel a (left). (**b**) Nucleotide and amino acid sequence identity for ADAMTS6, 10, 17, and 19 shows higher sequence identity for ADAMTS6 and ADAMTS10, and ADAMTS17 and ADAMTS19, respectively (pink). The following DNA/protein sequences from the NCBI database were used to calculate the percentage identity after sequence alignment with Clustal Omega (https://www.ebi.ac.uk/Tools/msa/clustalo/): ADAMTS6: NM_197941.4/NP_922932.2; ADAMTS10 NM_030957.4/NP_112219.4; ADAMTS17 XM_005254872.3/XP_005254929.1; ADAMTS19 NM_133638.6/NP_598377.3.

**Figure 2 biomolecules-10-00596-f002:**
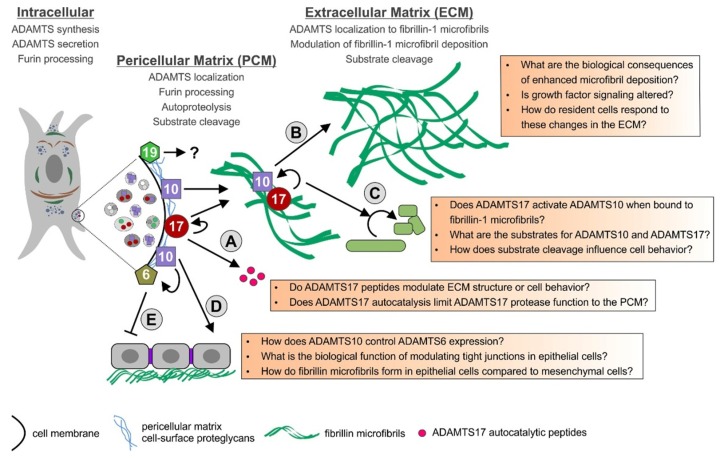
Model for the function of ADAMTS6, 10, 17, and 19 and their possible interactions. The ADAMTS proteases are furin-processed in the secretory pathway or at the cell surface and remain cell surface-associated or are released in the ECM. ADAMTS17 is autocatalytically processed and ADAMTS17 peptides are released from the cell surface (A). ADAMTS10 and ADAMTS17 bind to fibrillin microfibrils. ADAMTS10 can augment the deposition of fibrillin-1 microfibrils in the ECM (B). Fibrillin microfibrils could bring ADAMTS10 and ADAMTS17 together allowing ADAMTS17 to activate ADAMTS10 (C). ADAMTS10 also plays a role in regulating tight junctions and fibrillin microfibril formation in epithelial cells (D). ADAMTS10 can suppress ADAMTS6 gene expression through an unknown mechanism. When ADAMTS6 is overexpressed in epithelial cells, tight junctions, focal adhesions, and fibrillin microfibrils are disorganized (E). The role of ADAMTS19 in the PCM and ECM is currently unknown, but ADAMTS19 likely works as a true protease.

**Table 1 biomolecules-10-00596-t001:** Comparison of features distinguishing the individual ADAMTS6, 10, 17, and 19 proteases.

	ADAMTS6	ADAMTS10	ADAMTS17	ADAMTS19
N-Glycosylation sites **^1^**	6	6	7	5
O-Fucosylation sites **^2^**	3	3	4	4
Furin consensus sequences **^3^**	2	1	2	2
Furin processing **^4^**	yes	no	yes	yes
Autocatalysis	no	no	yes	no
Substrates	LTBP1, SDC4	FBN1, FBN2	ADAMTS17	n.d.
Alternative splicing	yes	yes	yes	likely
Human disorders	Prolonged QRS syndrome	WMS 1 (MIM #277600)	WMS 4 (MIM #613195)	Non-syndromic heart valve disease
Disease-causing human mutations	2	9	9	2
Gene knockout phenotype in mice	Prenatal/neo-natal lethality, double outlet right ventricle, ventricular hypertrophy, atrial and ventricular septal defects	Some prenatal/ neonatal lethality, abnormal ciliary zonule, shorter long bones due to growth plate abnormalities, skeletal muscle abnormalities	Some prenatal/ neonatal lethality, skeletal growth impairment due to growth plate abnormalities, brachydactyly by 8 mos. of age	Aortic valve dysfunction in ~40% of knockout mice
Fibrillin-1 binding	n.d.	yes	yes	n.d.
Fibrillin microfibril formation **^5^**	n.d.	Promotes FBN1 deposition	No effect	n.d.

**^1^** Consensus sequence for N-glycosylation: *Nx(no P)S/T*; **^2^** Consensus sequence for O-fucosylation: *Cxx(S/T)C*; **^3^** Consensus sequence for furin processing: *Rx(R/K)R*; **^4^** Determined by expression of recombinant ADAMTS proteases in HEK293 cells; **^5^** Determined in cell culture experiments; n.d., not determined.

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
