# Peer review of "The ADAMTS/Fibrillin Connection: Insights into the Biological Functions of ADAMTS10 and ADAMTS17 and Their Respective Sister Proteases"

_biomolecules, 2020, doi:10.3390/biom10040596_

Round 1
Reviewer 1 Report
Review by Karoulias et al. describes the relationships between fibrillin and the ADAMTS10 (and its closely related ADAMTS6) and ADAMTS17 (and its closely related ADAMTS19) metalloproteases. Overall this manuscript includes main findings concerning the interactions between ADAMTSs and fibrillin, which is important in the field.
In my opinion some aspects should be clarified or included:
- Lines 48-50. Authors refer to an almost ten years old article (reference 3) to mention the orphan ADAMTSs. I would recommend to include some more recent review or articles where a potential reader can consult new ADAMTSs substrates, even if those were “in vitro” substrates.
- Line 99. “Unpublished data” by the first author is included and then reference 24 is also indicated. In this part of the review authors are describing that both ADAMTS6 and ADAMTS19 are furin processed but do not undergo autocatalysis. Reference 24 is only for ADAMTS6 and does not mention something relevant in relation to autocatalysis. Could the authors clarify, please?.
- Line 348. “ADAMTS10 is an inactive protease”. This sentence does not sound right and can generate a lot of confusion. A sentence like that should be employed for certain ADAMs containing a catalytically inactive metalloprotease domain, but not in the case of the ADAMTSs. If enzymatic activity of a particular ADAMTS is modest, the best reaction conditions or the “in vivo” activity have not been yet found, it does not mean that is “inactive”. Without going any further, in line 236-237 is indicated that ADAMTS10 was shown to cleave fibrillin-2. Please remove or change that sentence.
- Line 367. “Mutations in ADAMTS19 were linked to a human disorder only very recently”. Reference should be cited.
Author Response
Reviewer 1:
Review by Karoulias et al. describes the relationships between fibrillin and the ADAMTS10 (and its closely related ADAMTS6) and ADAMTS17 (and its closely related ADAMTS19) metalloproteases. Overall this manuscript includes main findings concerning the interactions between ADAMTSs and fibrillin, which is important in the field. In my opinion some aspects should be clarified or included:
Lines 48-50. Authors refer to an almost ten years old article (reference 3) to mention the orphan ADAMTSs. I would recommend to include some more recent review or articles where a potential reader can consult new ADAMTSs substrates, even if those were “in vitro” substrates.
The reason, we chose that reference is that it contains a nice homology tree showing the different clades of ADAMTS proteases, which broadly overlap with the classes of substrates. We now included a couple of newer reference in line 49. In lines 50-52 we also mention five recent papers, published between 2017 and 2019 that are examples of a recent surge of studies that identified substrates for these proteases.
Line 99. “Unpublished data” by the first author is included and then reference 24 is also indicated. In this part of the review authors are describing that both ADAMTS6 and ADAMTS19 are furin processed but do not undergo autocatalysis. Reference 24 is only for ADAMTS6 and does not mention something relevant in relation to autocatalysis. Could the authors clarify, please?
In line 117-118 we now indicate that the statement is based on the western blot analysis shown for recombinant ADAMTS6 and we specify that our unpublished data refer to ADAMTS19. Figure 3A in the ADAMTS6 reference (Prins and Mead et al, 2018) shows a single 150 kDa band for ADAMTS6 in the lysate and a smaller, 130 kDa band in the medium. This would indicate that the full-length protein is secreted and processed, but not autocatalytically degraded. However, the active site mutant of ADAMTS6 was not tested and thus we cannot rule out the possibility that inactive ADAMTS6 could accumulate in larger quantities or that autocatalysis may occur in vivo. We included a respective comment in lines 118-120.
Line 348. “ADAMTS10 is an inactive protease”. This sentence does not sound right and can generate a lot of confusion. A sentence like that should be employed for certain ADAMs containing a catalytically inactive metalloprotease domain, but not in the case of the ADAMTSs. If enzymatic activity of a particular ADAMTS is modest, the best reaction conditions or the “in vivo” activity have not been yet found, it does not mean that is “inactive”. Without going any further, in line 236-237 is indicated that ADAMTS10 was shown to cleave fibrillin-2. Please remove or change that sentence.
We agree with the reviewer that this may be a misleading statement and we have rephrased the sentence to clarify that there is a possibility that ADAMTS10 has only weak or no protease activity if not activated by furin. But even weak protease activity could be relevant in slower matrix remodeling processes. We expanded the discussion of these possibilities in lines 317-340 and 399-407. We also indicated the possibility that optimal conditions for ADAMTS10 may only present in vivo, as suggested, or that substrates that are cleaved more efficiently by ADAMTS10 are not known yet.
Line 367. “Mutations in ADAMTS19 were linked to a human disorder only very recently”. Reference should be cited.
We included the reference at the end of the sentence in line 413.
Reviewer 2 Report
The review explores the connection of ADAMTS proteases to fibrillin microfibril biology and proposes a conceptual model describing the interaction of these ADAMTS proteases with the extracellular and pericellular matrix.The review is undoubtedly a significant contribution to the field. However the text is written in a way that correspond to the research strategy of the authors rather than as a message to be conveyed to large audience. The absence of continuity between chapters limits significantly the readability.The ADAMTS family comprises 19 secreted metalloproteases.The reason for focusing on a few of them is unclear.
On the opposite, the last chapter:Outlook:How may ADAMTS 6,10,17 and 19 cooperate in ECM formation and remodeling-describes very comprehensively the objectives of the review and proposes a series of challenging open questions (Figure 2) associated to the biological and pathological functions of some proteases pairs.The chapter opens the way to new research possibilities, includes the opinion of the authors about several aspects and contributes to strongly improve the impact of the review.
Author Response
Reviewer 2:
The review explores the connection of ADAMTS proteases to fibrillin microfibril biology and proposes a conceptual model describing the interaction of these ADAMTS proteases with the extracellular and pericellular matrix. The review is undoubtedly a significant contribution to the field. However, the text is written in a way that correspond to the research strategy of the authors rather than as a message to be conveyed to large audience. The absence of continuity between chapters limits significantly the readability.
We agree with the reviewer and in response we extensively revised the manuscript to connect the chapters and improve continuity. For example, we moved the paragraph describing alternative splicing (now lines 100-113) to better fit with the section on DNA and amino acid sequence analysis. In addition, we repositioned the paragraph describing the different mouse models to now follow the description of the human disorders (now lines 284-340). With that, the paragraph discussing the substrates and potential biological function is now directly followed by the “Outlook” section, which now provides a better flow and integration of the sections. Finally, we included several sentences connecting paragraphs and sections to improve the continuity and flow of the entire manuscript. We hope that the reviewer can appreciate that these changes now result in a much improved manuscript that is better suited for the broader readership of Biomolecules.
The ADAMTS family comprises 19 secreted metalloproteases. The reason for focusing on a few of them is unclear.
The main reason to focus on ADAMTS10 and ADAMTS17 and their sister proteases is that ADAMTS10 and ADAMTS17 are genetically linked to fibrillin-1, because mutations in these genes cause Weill-Marchesani syndrome. Biochemical and functional interactions between ADAMTS10, ADAMTS17 and FBN1 have been demonstrated as well. We decided to include the closest homologues for each of the proteases, ADAMTS6 and ADAMTS19, respectively, because ADAMTS proteases tend to form pairs and it is sometimes necessary to deplete both proteases to elicit or augment a phenotype in mice. We include a better rationale for the proteases in the abstract, lines 20-27, and at the end of the introduction section, lines 56-66, of the revised manuscript.
On the opposite, the last chapter: Outlook: How may ADAMTS 6,10,17 and 19 cooperate in ECM formation and remodeling-describes very comprehensively the objectives of the review and proposes a series of challenging open questions (Figure 2) associated to the biological and pathological functions of some proteases pairs. The chapter opens the way to new research possibilities, includes the opinion of the authors about several aspects and contributes to strongly improve the impact of the review.
We thank the reviewer for this very encouraging comment.
Round 2
Reviewer 2 Report
Most questions and suggestions have been addressed convincingly.The amended version of the manuscript has been improved significantly. The authors made a significant effort to improve the connection between chapters and the revised version is much better suited for a broader readership.